# New Chalcogenide Glass-Ceramics Based on Ge-Zn-Se for IR Applications

**DOI:** 10.3390/ma15145002

**Published:** 2022-07-18

**Authors:** Alin Velea, Florinel Sava, Petre Badica, Mihail Burdusel, Claudia Mihai, Aurelian-Catalin Galca, Elena Matei, Angel-Theodor Buruiana, Outman El Khouja, Laurent Calvez

**Affiliations:** 1National Institute of Materials Physics, Atomistilor 405A, 077125 Magurele, Romania; alin.velea@infim.ro (A.V.); petre.badica@infim.ro (P.B.); mihai.burdusel@infim.ro (M.B.); claudia.mihai@infim.ro (C.M.); ac_galca@infim.ro (A.-C.G.); elena.matei@infim.ro (E.M.); angel.buruiana@infim.ro (A.-T.B.); outman.elkhouja@infim.ro (O.E.K.); 2Faculty of Physics, University of Bucharest, Atomistilor 405, 077125 Magurele, Romania; 3ISCR (Institut des Sciences Chimiques de Rennes) UMR 6226, Université de Rennes, CNRS, F35000 Rennes, France; laurent.calvez@univ-rennes1.fr

**Keywords:** glass-ceramics, chalcogenide, IR optical elements

## Abstract

The consumer market requests infrared (IR) optical components, made of relatively abundant and environmentally friendly materials, to be integrated or attached to smartphones. For this purpose, three new chalcogenides samples, namely Ge_23.3_Zn_30.0_Se_46.7_ (d_GZSe-1), Ge_26.7_Zn_20.0_Se_53.3_ (d_GZSe-2) and Ba_4.0_Ge_12.0_Zn_17.0_Se_59.0_I_8.0_ (d_GZSe-3) were obtained by mechanical alloying and processed by spark plasma sintering into dense bulk disks. Obtaining a completely amorphous and homogeneous material proved to be difficult. d_GZSe-2 and d_GZSe-3 are glass-ceramics with the amount of the amorphous phase being 19.7 and 51.4 wt. %, while d_GZSe-1 is fully polycrystalline. Doping with barium and iodine preserves the amorphous phase formed by milling and lowers the sintering temperature from 350 °C to 200 °C. The main crystalline phase in all of the prepared samples is cubic ZnSe or cubic Zn_0.5_Ge_0.25_Se, while in d_GZSe-3 the amorphous phase contains GeSe4 clusters. The color of the first two sintered samples is black (the band gap values are 0.42 and 0.79 eV), while d_GZSe-3 is red (E_g_ is 1.37 eV) and is transparent in IR domain. These results are promising for future research in IR materials and thin films.

## 1. Introduction

Infrared (IR) radiation is that part of electromagnetic spectrum between the visible and microwave radiation (with the wavelength between 0.78 and 1000 µm). The driving force for the development of IR transparent materials was initially related to military applications. Today, the IR devices using optical elements made of IR transparent materials belong to the consumer market. They can be integrated or attached to smartphones [1,2] and are used in automotive industry for driving assistance during night time. These IR devices have two functionalities: (i) night vision of bodies or objects at room temperature which emit IR radiation with the wavelength between 7 and 4 μm and (ii) thermovision of warm bodies or objects which emit IR radiation with the wavelength between 3 and 5 μm. The second functionality is also very useful to detect patients with fever due infectious diseases, which can be easily transmitted leading to outbreaks.

The most important requirements for the IR transparent materials are the following: high transmittance that covers also the range of longer wavelengths, a high degree of optical homogeneity and high purity, appropriate refractive index and dispersion parameters in the infrared region, thermal stability against crystallization (resistance to devitrification), and durability in air and against water or mechanical stress [3].

The accessibility of IR optical elements on the consumer market is due to the fact that IR transparent chalcogenide glasses and glass-ceramics allow for easy production by molding [4,5], while their weak mechanical properties are no longer a major impediment. However, the stability against crystallization remains the main constraint, in addition to the extended transparency across the entire visible spectrum, which is the ultimate goal that has not been achieved so far.

The IR transparent chalcogenide glasses and glass-ceramics should be composed of relatively abundant, nontoxic, and environmentally friendly elements. The Ge-Zn-Se system meets these requirements. Moreover, the presence of crystalline Ge and ZnSe in the Ge-Zn-Se glass-ceramics, that have a very good IR transparency (up to 16 µm [6]), is an advantage. Furthermore, both Ge and Zn have a tetrahedral coordination which, if preserved in the amorphous phase, will produce good stability against crystallization. Also, this system can be doped with barium and iodine. The addition of heavy atoms (such as barium) in IR chalcogenide glasses and glass-ceramics decreases the optical bandgap [7] and could increase the long-wavelength transmission edge [3]. On the other hand, the addition of halogen iodine decreases the amount of dangling bonds in the glass network and the average coordination number of the atoms, leading to the stabilization of the disordered structure against crystallization. It also increases the optical bandgap producing an increase in the transparency towards/into the visible spectrum [8,9]. 

The glassy Ge-Zn-Se system was very little studied (to the best of our knowledge). Vassilev et al., using the melt quenching technique, determined the glass-formation domain of Ge-Zn-Se system [10] and the maximum solubility of Zn in the glassy Ge-Se network, which is about 5% at a Ge:Se ratio of 9:1. Choi et al. succeeded to increase the solubility of Zn in RF sputtered amorphous Ge-Se thin films up to 23% [11], and proved that Zn is four-fold coordinated and it bonds preferentially to Se atoms. de Lima et al. [12], using the ball milling technique, have obtained only cubic zinc selenide, zinc, and amorphous selenium phases in the chalcogenide Zn-Se system. This result confirms that ZnSe is the only chemical compound that can be obtained in the binary Zn-Se phase diagram [13]. Finally, in the orange-yellow BaZnGeSe_4_ crystals, obtained by Yin et al. [14], Ge and Zn atoms are four-fold coordinated and form one-dimensional chains which are separated by Ba cation chains.

The objective of this work is to produce glass-ceramics from the Ge-Zn-Se system, with the Zn concentration greater than the 5% limit, undoped or doped with Ba and I and transparent in the IR domain. For this purpose, a new preparation route involving mechanical alloying and SPS, is explored. New chalcogenide compositions, namely Ge_23.3_Zn_30.0_Se_46.7_, Ge_26.7_Zn_20.0_Se_53.3_, and Ba_4.0_Ge_12.0_Zn_17.0_Se_59.0_I_8.0_ were obtained by ball milling. Subsequently, the powders were processed by Spark Plasma Sintering (SPS) to produce the bulk glass-ceramics disks. The structural evolution of the bulk materials was characterized using both grazing incidence X-ray diffraction (GIXRD) and Raman spectroscopy and compared with the precursor powders. while its morphological changes were observed using both optical and scanning electron microscopy. Moreover, the optical properties of the obtained materials were investigated by spectroscopic ellipsometry and Fourier transform infrared (FTIR) spectroscopy. 

## 2. Materials and Methods

The used raw materials were GeSe_2_ (MuseChem, Fairfield, NJ, USA, purity 99.9%), Zn (Sigma-Aldrich, Burlington, MA, USA, purity 99.0%), Se (Fluka AG, Buchs, Switzerland, 99.9%), and BaI_2_ (MuseChem, Fairfield, NJ, USA, purity 99.0%) powders. Milling was employed for mechanical alloying in a Retsch PM400 (Haan, Germany) planetary ball mill. The prepared compositions given in at. % were Ge_23.3_Zn_30_Se_46.7_ (denoted GZSe-1), Ge_26.7_Zn_20_Se_53.3_ (GZSe-2) and Ba_4.0_Ge_12.0_Zn_17.0_Se_59.0_I_8.0_ (GZSe-3). The preparation details are summarized in Table 1. The weighed powders were introduced in a 125 mL tungsten carbide (WC) grinding jar together with WC milling balls of different sizes. Rotation cycles of 3 min at 400 rpm were scheduled with direction reversal and a pause of 3 or 6 min between each cycle. The milling atmosphere was Ar, with and oxygen mole fraction of less than 0.3%.

The first two compositions, GZSe-1 and GZSe-2, were obtained within a one-step milling process using the indicated raw materials. In the case of GZSe-3, a three-stage process was employed. First, ZnSe was prepared. This material was milled in the second stage with BaI_2_ and Se to produce a precursor powder with the composition Ba_6.2_Zn_26.6_Se_54.7_I_12.5_. In the third stage the previously prepared precursor powder was mixed with GeSe_2_ and milled again to obtain the Ba_4.0_Ge_12.0_Zn_17.0_Se_59.0_I_8.0_ (GZSe-3) composition.

Samples of several tens of milligrams of the powder obtained by mechanical alloying were extracted at certain intervals of time. Each sample was placed on a silicon holder as a compact layer and was investigated by grazing incidence X-ray diffraction. A SmartLab diffractometer (Cu K_α_ radiation, λ = 1.54178 Å, Rigaku, Japan) equipped with HyPix-3000 2D Hybrid Pixel Array Detector (Rigaku, Tokyo, Japan) in 0D mode was used.

The milled powders with the three compositions were densified by spark plasma sintering, using a FCT Systeme GmbH, HP 5D equipment, to obtain bulk disks. Sintered bulk materials were labelled d_GZSe-1, d_GZSe-2, d_GZSe-3. For SPS processing typical graphite dies (with a diameter of 15 mm) and punches were used. In order to prevent carbon diffusion in the disks, which can affect the transparency in IR, the powders were wrapped into tantalum foils. Between the tantalum foils and the graphite die, graphite foils were introduced to ensure easy sliding of the moving parts. 

In Figure 1 are shown the processing parameters during SPS. The vacuum pressure in the working chamber of SPS furnace was 36 Pa. The heat treatment during SPS was designed based on preliminary experiments aiming to preserve the amorphous phase of the milled powders and to avoid evaporation of volatile components, but at the same time to produce a dense sintered material. In this respect, the initial regime was of non-equilibrium, where a strong electric current is applied for a short period of time, between 15 to 40 s (which corresponds to a sudden increase in temperature in Figure 1), necessary to initiate disk compression. Then, the SPS temperature was constant: 350 °C, for d_GZSe-1 and d_GZSe-2, and 200 °C for d_GZSe-3.

The sintered disks were mirror polished carefully on both sides using increasingly fine-grained emery sheets (grade up to 1500) and felt with diamond paste (granulation 0.5 and then 0.25 μm). The structure of the mirror polished bulk disks was also investigated by GIXRD using the same diffractometer with the detector in 1D mode. The crystalline phases have been identified using the DIFFRAC.SUITE Software package (Bruker, Billerica, MA, USA). The average size of the crystallites was estimated by using the Scherrer equation. A standard sample of Al_2_O_3_ was measured and used to eliminate the instrumental broadening of the crystalline peaks width.

The morphology of the discs was examined using with a Zeiss EVO 50 XVP (Carl Zeiss AG, Oberkochen, Germany) scanning electron microscope (SEM). Local chemical composition was measured by energy-dispersive spectrometry (EDS) (Bruker, Billerica, MA, USA).

Raman spectra were recorded at room temperature, in the 50–400 cm^−1^ range, in backscattering configuration, with a LabRAM HR Evolution spectrometer (Horiba Jobin-Yvon, Palaiseau, France) equipped with a confocal microscope. A He–Ne laser (Horiba Jobin-Yvon, Palaiseau, France) operating at 325 and 633 nm was focused using an Olympus 100× objective (Olympus, Tokyo, Japan) on the surface of the samples. The laser excitation power was adjusted to avoid laser-induced heating. Accurate and automated calibration was performed on a standard Si wafer (provided by Horiba Jobin-Yvon, Palaiseau, France) by checking the Rayleigh and Raman signals.

The wavelength dependence of the linear refractive indices (n), of the extinction coefficient (k), of the absorption coefficient (*α*) and of the ε_1_ and ε_2_ dielectric functions were measured on the polished glass-ceramics disks with a Variable Angle Spectroscopic Ellipsometer (J.A. Woollam Co., Lincoln, NE, USA), that incorporate a high-pressure Xenon discharge lamp (Hamamatsu Photonics K.K., Hamamatsu, Japan), and a HS-190 monochromator (J.A. Woollam Co., Lincoln, NE, USA).

The transmission spectra of the double-side mirror-polished glass-ceramic discs and also of different weight concentrations of grinded d_GZSe-3 sample embedded in a KBr matrix, were recorded by a Fourier transform infrared spectrophotometer (FTIR-6600, Jasco, Easton, MD, USA) in the range from 1.3–20 μm, in air.

## 3. Results

### 3.1. XRD Measurements

#### 3.1.1. Raw and Milled Powders

GeSe_2_ and Zn raw powders were measured by GIXRD to observe the phase type and purity (data not shown). They are crystalline and composed of a single phase. GeSe_2_ has a monoclinic crystal lattice, symmetry group P21/c (14) (PDF 04-003-1033), while Zn has a hexagonal crystal lattice (h-Zn), symmetry group P63/mmc (194) (PDF 00-004-0831).

The GIXRD patterns, measured on powder samples with composition GZSe-1 (Ge_23.1_Zn_26.7_Se_50.2_, wt. %, Table 1) and obtained for different milling times, are presented in Figure 2a. After 3 h of milling (Figure 2a, pattern 3 h) an amorphous phase forms. Its presence is inferred from the occurrence of the broad maxima at 14.090°, 28.570° and 51.090° respectively. The first broad maximum at 14.090°, the so-called first sharp diffraction peak (FSDP) is the signature of the medium-range order in the amorphous phase. This amorphous state is characterized by a less dense amorphous network [15] compared to a tetrahedral amorphous network (as e.g., for amorphous germanium), which is more compact and does not show a FSDP in the diffraction pattern. The monoclinic GeSe_2_ has a layered structure [16] with packing of quasi-2D layers, in which Germanium is tetrahedrally coordinated. When it is fully amorphous, it shows a FSDP at 14.090° (2θ) in the diffraction pattern [17]. Zinc is highly reactive with selenium from GeSe_2_ and, after 3 h of milling, forms a polycrystalline ZnSe cubic phase (c-ZnSe, symmetry group F-43m (216), PDF 00-037-1463). The amount of newly formed c-ZnSe is 19.8 wt. %, while 15.3 wt. % of unreacted hexagonal Zn (h-Zn) remains in the sample. A part of the selenium-depleted Ge-Se network crystallizes in the orthorhombic GeSe phase (o-GeSe, symmetry group Pcmn (62), PDF 04-004-3590). The amount of this phase was estimated at 7 wt. %. The structure of o-GeSe consists in a packing of quasi-2D layers, in which Germanium and Selenium are approximately octahedrally coordinated. The difference to 100 wt. %, i.e., 57.9 wt. %, is the disordered amorphous phase Ge_19.75_Zn_2.43_Se_35.72_ (wt. %). This result indicates that the amorphous Ge-Zn-Se alloy has begun to form.

After 21 h of milling, the unreacted h-Zn is only 0.1 wt. % and the c-ZnSe and o-GeSe phases increase to 58.7 wt. % and 13.4 wt. %, respectively. Consequently, the composition of the amorphous phase is Ge_16.7_Se_11.1_ (27.8 wt. %), where the Se content is less than in GeSe. Thus, the amorphous network is more compact (there are many Ge-Ge bonds) and this is the cause for the absence of the FSDP in the diffraction pattern (Figure 2a, pattern 21 h).

The same ratios are maintained after 150 h of milling (Figure 2a, pattern 150 h). However, after 242 h (Figure 2a, pattern 242 h), when the pause between the milling cycles has been halved from 6 to 3 min, the only change is the appearance of an unidentified crystalline phase (0.5 wt. %, indicated with “*” in Figure 2a pattern 242 h). This phase could be a distorted hexagonal ZnSe (h-ZnSe), as it is suggested by the relatively good match with the PDF 01-089-2940 (symmetry group P63mc (186)). The value of the cubic ZnSe lattice parameter (*a*) is 5.665 Å close to that in PDF 00-037-1463 (5.66882 Å). The final chemical composition, Ge_16.7_Se_11.0_ (where coefficients are in wt. %), of the amorphous phase from the GZSe-1 powder is denoted “a-P1”. After 242 h of milling, the color of the powder is black. 

The GIXRD patterns of GZSe-2 (Ge_26.0_Zn_17.5_Se_56.5_, wt. %, Table 1), taken every half hour up until 3 h of milling and again at 18 h, are presented in Figure 2b. After 0.5 h of milling (Figure 2b, pattern 0.5 h), the GeSe_2_ component becomes completely amorphous with a FSDP at 14.200° and the c-ZnSe phase formation is initiated. The weight percentage of c-ZnSe increases during the 3 h of milling, while the o-GeSe phase is still not formed. The phase composition in GZSe-2, after 3 h of milling is of 20.4 wt. % c-ZnSe, 8.3 wt. % unreacted h-Zn, and 71.3 wt. % amorphous phase, with the computed composition Ge_26.0_Se_45.3_ (where coefficients are in wt. %). 

After 18 h of milling, the amount of unreacted h-Zn is of only 0.1 wt. %, c-ZnSe (lattice parameter *a* = 5.661 Å is close to 5.66882 Å in PDF 00-037-1463) increases to 38.4 wt. %, and for the o-GeSe phase is 10.1 wt. %. The amount of amorphous phase, with the estimated composition of Ge_21.2_Se_30.2_, denoted “a-P2”, is 51.4 wt. %. In this amorphous phase (GeSe_1.42_), the Se content is higher than in GeSe and this is the reason for the presence of the FSDP at 14.050° in the diffraction pattern. The color of the powder milled for 18 h is black.

For the synthesis of the GZSe-3 composition samples, as addressed in Section 2, a precursor Zn-Se powder was obtained by ball milling (Table 1, stage 1). After 24 h of milling, the reaction between Zn and Se was completed and the powder precursor is a single-phase c-ZnSe material (Figure 2c, black curve stage 1—24 h). The cubic lattice parameter *a* is 5.667 Å, and the average crystallite size, D, (inferred from the width of the (111) peak) is 8.3 nm. The color of the ZnSe powder is black. 

In the second stage, milling for 24 h (Table 1) of the c-ZnSe precursor, with BaI_2_ and Se, produced a powder in which unreacted BaI_2_ and Se crystalline phases are no longer present, while the c-ZnSe phase still exists (Figure 2c, blue curve stage 2—24h). In addition, in the XRD pattern there are diffraction lines (indicated with “*” in Figure 2c pattern Stage 2—24 h) of an unidentified polycrystalline phase considering the ICDD database. Also, an amorphous phase, whose composition cannot be determined, is present. With the increase of the grinding time up to 169 h (Figure 2c, blue curve stage 2—169 h), the total amount of c-ZnSe (*a* increases to 5.669 Å, D = 8.8 nm) and of the unidentified polycrystalline phase decrease with 29 wt. %. This decrease is accompanied by an increase in the amount of the amorphous phase. There is no FSDP in the GIXRD patterns for stage 2 and the color of the powder after 169 h of milling is dark red.

In the third milling stage, the GeSe_2_ raw powder is added (see Table 1) to the previous synthesized powder. After 24 h of milling, the GZSe-3 (Ba_6.7_Ge_10.6_Zn_13.5_Se_56.8_I_12.4_, wt. %) alloyed powder is obtained. The lattice parameter of the resulted cubic polycrystalline phase has a lower value (*a* = 5.639 Å) than that of the standard c- (*a* = 5.66882 Å according PDF 00-037-1463) and it is similar with the value for the cubic c-Zn_0.5_Ge_0.25_Se (*a* = 5.646 Å, F-43m (216), PDF 04-017-7583). Therefore, the initial Zn-Se cubic phase could have incorporated Ge and contain cation vacancies. The GIXRD pattern (Figure 2c, red curve stage 3—24 h) shows that the unidentified polycrystalline phase and c-Zn_0.5_Ge_0.25_Se (D = 13.4 nm) dissolve in the amorphous matrix in an amount of 60 wt. % and 47 wt. %, respectively. There is no FSDP, which is a clear indication that the amorphous GeSe_2_ phase has chemically reacted with the other elements to form a new amorphous phase. This new amorphous phase has the average coordination of its elements higher than that of the typical amorphous GeSe_2_ (a-GeSe_2_). The phase composition of the GZSe-3 sample after 24 h of milling in stage 3 of 34.6 wt. % c-Zn_0.5_Ge_0.25_Se, 14.9 wt. % of the unidentified polycrystalline phase, and 50.5 wt. % of the new amorphous phase whose composition (denoted “a-P3”) could not be determined from the GIXRD measurement. The color of this powder is red, but it is lighter than the powder obtained after 169 h in the second milling stage.

#### 3.1.2. Bulk Sintered Disks

XRD patterns of the sintered chalcogenide samples (d_GZSe-1, d_GZSe-2, and d_GZSe-3) are exhibited in Figure 3, where optical images of the polished disk fragments are also shown. Fragmentation occurred in the thinning/polishing process suggesting the presence in the bulk samples of residual strain and/or of aging processes. Considering this detail, the XRD measurements were repeated on the bulk samples stored for eight months in the ambient atmosphere and the results are presented in Section 3.1.3.

The sintering conditions (see Figure 1), needed to densify the GZSe-1 powder, drastically influence the “a-P1” amorphous phase (Ge_16.7_Se_11.0_). This amorphous phase crystallizes as o-GeSe (thus the amount of this phase increases from 13.4 wt. % to 34.6 wt. %) (Figure 3a). The remaining germanium (6.5 wt. %) is in crystalline state, but its XRD peaks cannot be discriminated against the peaks of c-ZnSe, since the values of the cubic lattice parameters for both phases are very close: 5.66882 Å for c-ZnSe (according PDF 00-037-1463), vs. 5.6576 Å for c-Ge (according PDF 00-004-0545). The peaks assigned to c-ZnSe or c-Ge are very broad and a deconvolution cannot be performed. The amount of c-ZnSe (*a* increases to 5.668 Å, D = 33.1 nm) is 57.4 wt. %, while the unidentified phase (which could be a distorted hexagonal ZnSe, PDF 01-089-2940, P63mc (186)) increases to 1.5 wt. %. The color of the disk is black. 

In the case of the d_GZSe-2 sample, the percentages of the polycrystalline phases also show an increase. The amount of o-GeSe increases from 10.1 wt. % (in powder) to 37.3 wt. %, while the cubic phase enhances only with 4.6 wt. % on the account of Germanium that succeeds to enter in the ZnSe lattice. The cubic lattice parameter (*a*) decreases from 5.661 Å (in GZSe-2 powder) to 5.652 Å, closer to the value of c-Zn_0.5_Ge_0.25_Se (Zn_25.2_Ge_14.0_Se_60.8_) with *a* = 5.646 Å (PDF 04-017-7583), due to cation vacancies. However, the amorphous phase remains in a significant percentage (19.7 wt. %). The FSDP disappears in the XRD diffractogram, which means that the amorphous network is more compact now. This could be due to its composition, a-Ge_2.1_Zn_5.2_Se_12.4_ (denoted “a-D2”), rich in zinc, while Selenium probably remains tetrahedrally coordinated. The color of the disk is black.

The mild sintering conditions (see Figure 1), needed to densify the GZSe-3 powder, increased slightly the amount of amorphous phase (denoted “a-D3”), from 50.5 wt. % to 51.4 wt. %. The c-Zn_0.5_Ge_0.25_Se (Zn_25.2_Ge_14.0_Se_60.8_, where coefficients are in wt. %) phase still exists (experimental value *a* = 5.648 Å) and increases from 34.6 wt. % to 43.4 wt. %, while the unidentified polycrystalline phase (or phases) disappears and two new unidentified polycrystalline phases are formed (5.2 wt. % total amount). The color of the disk is red. 

The weight percentages of the phases in each sample (powder or sintered materials) are presented in Table 2. 

#### 3.1.3. Time Stability (Aging) of the Sintered Materials

After 8 months, the average crystallite size of the d_GZSe-1 sample increased with 74.3% for c-ZnSe and with 125% for o-GeSe (Table 3), but the phases and their amount have remained the same. 

In the case of d_GZSe-2, the c-Zn_0.5_Ge_0.25_Se phase transforms into c-ZnSe (*a* increases to 5.662 Å). The amorphous phase is also transformed since the FSDP appears at 13.610° (2θ), therefore the phase becomes less dense and the average coordination per atom decreases. The average size of the crystallites increases with 33% for c-ZnSe and with 109% for o-GeSe (Table 3). 

The percentage of the cation vacancies in the c-Zn_0.5_Ge_0.25_Se phase is reduced in the case of the d_GZSe-3 sample, thus the cubic lattice parameter (*a*) increases from 5.648 Å to 5.657 Å (*a* for c-ZnSe is 5.66882 Å, according PDF 00-037-1463). The amorphous phase also is transformed since the FSDP appears at 15.15° (2θ), as in the previous case, and it becomes less dense. The average size of the crystallites for the cubic ZnSe phase increases with 12% (Table 3). 

### 3.2. SEM and EDS

The SEM images, at different magnifications (5000×, 10,000×, and 20,000×) of the cross section of the sintered disks d-GZSe-1, d_GZSe-2, d_GZSe-3, are shown in Figure 4.

The highest roughness and porosity are for the sample d_GZSe-3, while the lowest are for the sample d_GZSe-2 and intermediate values are found for d_GZSe-1. The low porosity in sample d_GZSe-2 might be associated with the presence of a significant amount of amorphous phase (19.7 wt. %) that can conveniently fill in the spaces between crystalline grains. 

The average EDS compositions of the samples, measured at the smallest magnification (5000×), are presented in Table 4. In general, the samples can be considered homogeneous. The measured EDS compositions of the bulks match within 5 at. % with those calculated for the fabrication of the powder mixtures by milling. 

### 3.3. Raman Spectroscopy

The micro-Raman spectra, recorded on the three mirror-polished chalcogenide bulk samples, are shown in Figure 5. The wavelength of the monochromatic incident light was 325 nm (3.815 eV, left) and 633 nm (1.959 eV, right), respectively. It was found that, at high laser power, there are changes of the irradiated area (photo-structural transformations that are specific to chalcogenide materials [18]), and, therefore, only the spectra at lower laser power with minimal influence on the samples, are shown. This is also the reason for the high noise in the spectra of d_GZSe-1 and d_GZSe-2, obtained at 325 nm.

The Raman peaks positions for the three bulk samples are presented in Table 5. For comparison and phase identification, in Table 6 are indicated the peaks positions for different phases as found in literature.

The d_GZSe-1 and d_GZSe-2 chalcogenide samples have similar Raman spectra for both wavelengths (Figure 5). For the c-ZnSe/c-Zn_0.5_Ge_0.25_Se peak (252/254 nm, respectively), the longitudinal optical phonon mode is evidenced only by irradiation with 325 nm, while the o-GeSe peaks are better observed for irradiation with 633 nm. The presence of the c-Ge phase in d_GZSe-1 is evidenced when irradiated with 633 nm. The Raman measurements confirm the XRD results. Two peaks, at 76 and 100 cm^−1^, could not be assigned to a known phase.

For the d_GZSe-3 bulk sample, only the amorphous phase was evidenced in the Raman spectra for both wavelengths of the excitation radiation (Figure 5). The large width of the peaks indicates a disordered/amorphous phase. Composition of this phase could not be inferred from the XRD analysis, but the Raman spectra suggest that it contains as structural building blocks the Ge-centered tetrahedra, GeSe_4/2_. This block can be connected in two simple ways, by sharing a single corner (named corner-sharing, CS, GeSe_4/2_ tetrahedra) or an edge (named edge-sharing, ES, Ge_2_Se_8/2_ bi-tetrahedra). Another structural building block of the amorphous phase is the polymeric Se_n_ chain. The position and intensity of the CS, ES, and Se_n_ peaks in the Raman spectra depend on the ratio of Ge/Se percentage in the a-Ge_x_Se_1-x_ phase. Thus, the broad Raman peaks for d_GZSe-3 are very close to those of the a-GeSe_4_ (Ge_18.7_Se_81.3_), as shown in Table 6. Namely, four peaks are very well distinguished (the noise in the spectra is very low) at 195 cm^−1^ (CS, A_1_ vibrations, the symmetric breathing motions of chalcogen atoms at GeSe_4/2_ tetrahedra), 211 cm^−1^ (ES, the companion A_1_^c^ peak), 298 cm^−1^ (ES), and at 257 cm^−1^ (the symmetric stretch of polymeric Se_n_ chains) [24]. Therefore, the Raman analysis shows that Germanium enters only in the amorphous phase and probably in c-Zn_0.5_Ge_0.25_Se. Accordingly, the amorphous phase contains at least 23.5 wt. % GeSe_4_ clusters, the rest until 51.4% being the contribution of the other elements, which do not have a Raman signature. The unidentified crystalline phase and c-Zn_0.5_Ge_0.25_Se are not revealed in the Raman spectra.

### 3.4. Spectroscopic Ellipsometry

The optical properties of the three sintered bulk samples were analyzed using an ellipsometer in the wavelength range from 250 to 1750 nm. The refractive index (n), extinction coefficient (k) and absorption coefficient (α = 4πk/λ, λ is the wavelength) are presented in Figure 6a. The dielectric functions directly determined from ellipsometry are shown in Figure 6b. The sensitivity for very small extinction coefficients is lower for ellipsometry than in the case of conventional transmission spectroscopy [26], but due to sample properties (high electron density) and processing issues (porosity), conventional transmittance was not possible to be performed.

The bandgap (E_g_) of sintered samples was calculated using the equation (αhν)^n^ = A(hν − E_g_), where A is a constant, h is Planck’s constant, ν is the frequency of the incident photon, and n equals to 1/2 for materials with an indirect band gap. The E_g_ values obtained are 0.42 eV, 0.79 eV, and 1.37 eV for d_GZSe-1, d_GZSe-2, and d_GZSe-3, respectively. The indirect band gap for o-GeSe is 1.14 eV [27], for c-ZnSe is 2.68 eV [28], for c-Ge is 0.661 eV [29], while for a-GeSe_4_ is 2.07 eV [24]. The computed band gaps of the synthesized discs are lower than the experimental/theoretical values due to the mixture of crystalline/amorphous phases. The d_GZSe-3 sample has the largest bandgap since it has the highest quantity of amorphous phase and probably it contains barium atoms that decrease the optical bandgap of a-GeSe_4_ [7]. Light scattering due to embedded crystals might also alter the values.

For the d_GZSe-1 sample the dominant crystalline phase is c-ZnSe. The values of ε_1_ and ε_2_ (Figure 6b) match relatively well to those previously reported for ZnSe [30,31], but there are also some discrepancies. There are several factors that can influence the optical properties measured on the sintered discs, such as specifics of processing and sample type (e.g., a thin film or a massive bulk). Our results are comparable with those obtained on films with a thickness larger than 200 nm (critical thickness). Above this value, the film can be considered as having bulk properties [28]. Three main critical points, namely E_0_, E_0_ + Δ_0_ and E_1_, can be distinguished in ε_2_. The energies of these points are 2.68 eV, 3.11 eV and 4.55 eV, respectively. Except E_1_, these values are similar with the findings of M. Cardona [30], i.e., 2.67 eV, 3.10 eV, and 4.7 eV, respectively. In our case, the peak centered at 4.55 eV is very broad. This can be due to the fact that GeSe also has a transition at 4.3 eV [32]. Moreover, crystalline Ge also shows the E_2_ peak at 4.23 eV. Since the sample d_GZSe-1 contains significant quantities of o-GeSe and c-Ge, the broad peak at 4.55 eV can be a superposition of those signals. On the other hand, in d_GZSe-2, E_1_ is displaced at 4.7 eV, which is the exact value reported by M. Cardona [30], suggesting that the broadness of the peak from d_GZSe-1 is caused by the presence of c-Ge which is not present in d_GZSe-2. The sample d_GZSe-3 has an amorphous-like dielectric function without any sharp features, being in good agreement with the XRD and Raman results.

### 3.5. FTIR Spectroscopy

Attempts were made to measure the IR transmission of polished glass-ceramics discs, however, the non-parallel surfaces and the samples volume gave inaccurate values. Next, different weight concentrations (1.0, 2.5, and 5.0 wt. %) of the grinded d_GZSe-3 sample were measured in a KBr matrix. To avoid the effects due to mechanical attrition, the embedded particles are of several microns in size. They, therefore, light scattering is induced (in Figure 7, notice the increase of the transmittance for longer wavelengths), which reduces the transparency window. Two absorption bands are noticeable at 12.78 μm and 17.89 μm which are due to the oxidation of the chalcogenide particles surfaces [33]. Further research is required to assess in detail the optical IR properties (e.g., transmittance). 

## 4. Discussion

The aim of this work was to obtain and explore new homogeneous amorphous Ge-Zn-Se alloys, undoped and doped with Ba and I, targeting their use for IR optical devices. Our experiments revealed some difficulties worth discussing. 

One problem was the strong and the fast tendency of Zinc to form with selenium, taken out of GeSe_2_, the stoichiometric and very stable crystalline ZnSe phase. If Ge and Se powders would have been used as raw materials, the reaction of Zinc with selenium would be even more favored due to the long alloying time between germanium and Selenium [33]. The result is in agreement with other studies showing that once formed, ZnSe cannot be amorphized by milling [34,35]. This fact is likely due to the value of the torsional force constant (the torsion being the deviation of the dihedral angle from the equilibrium values corresponding to a symmetry of order n), which imposes tetrahedral coordination for Selenium. This value seems to be higher than that in amorphous Ge-Se alloys or in o-GeSe_2_, where the coordination of Selenium is 2 [11,16].

The second difficulty was that o-GeSe, once formed (the germanium and selenium are threefold-coordinated), is very difficult to be amorphized by milling. It does not become completely amorphous even after 300 h of milling [17]. Also, the crystalline germanium powder transforms into amorphous germanium only 15–18% of the total powder volume [36] by ball milling.

These energetic barriers work against the fabrication of a homogeneous Ge-Zn-Se alloy by ball milling. However, a demonstrated solution to overcome these barriers is the reaction between the BaSe, ZnSe, Ge and Se powders at 1173 K to obtain orange-yellow crystals of BaZnGeSe_4_ [14].

In our case, the addition of barium and iodine in the Ge-Zn-Se alloy proved to be very useful. Thus, the “a-P3” amorphous phase (50.5 wt. % in the milled GZSe-3 powder), increased in the SPS process to 51.4 wt. % (“a-D3” in the d_GZSe-3 sample), while, in the undoped composition, “a-P1” (27.7 wt. % in the milled powder GZSe-1) and “a-P2” (51.4 % in the milled powder GZSe-2) drastically decreased during sintering (in the d_GZSe-1 and d_GZSe-2 samples, respectively). The sintered d_GZSe-1 sample is fully crystalline and “a-D2” is only 19.7 wt. % in d_GZSe-2. Addition of the indicated elements also decreased the SPS processing temperature.

The GZSe-3 powder was obtained at the end of three successive sintering stages, and the color of the powders obtained at the end of these stages is intriguing. Thus, after the first stage, the ZnSe powder is black, a result also obtained in other works [35]. This shows that the surface of the ZnSe nanometric sized particles is very defective, and given the large number of surface atoms relative to the volume atoms [35], these defects drastically decrease the value of the bandgap, leading to a dark colored powder. At the end of the second stage, the addition of BaI_2_ and Se to ZnSe produced a dark red Ba_6.2_Zn_26.6_Se_54.7_I_12.5_ powder. Finally, the incorporation of GeSe_2_ causes the final GZSe-3 powder to become light red. This might be due to the formation of the amorphous GeSe_4_ (which is red [33]) in the GZSe-3 sample, as the Raman Spectroscopy revealed. Further sintering by SPS did not produce any changes in the color of the d_GZSe-3 bulk disc.

Another aspect to be considered is that thin films may promote the required transparency of materials fabricated in this work as addressed in Section 3.5. In the literature, amorphous Ge_53_Zn_23_Se_24_ films prepared by radio-frequency sputtering at room temperature [11] were reported. This is promising, but further research is needed.

## 5. Conclusions

Two new chalcogenide glass-ceramics in the Ge-Zn-Se system, undoped and doped with Ba and I, have been successfully prepared by mechanical alloying and spark plasma sintering. The amount of the amorphous phase in both glass-ceramics, namely Ge_26.7_Zn_20.0_Se_53.3_ and Ba_4.0_Ge_12.0_Zn_17.0_Se_59.0_I_8.0_ is 19.7 and 51.4 wt. %, while the Zn content is greater than the 5% solubility limit. a-GeSe_4_ clusters are present in an important quantity in GZSe-3. The average crystallite size is nanometric in the prepared samples. The color of d_GZSe-2 is black (E_g_ = 0.79 eV), while d_GZSe-3 is light red (E_g_ is 1.37 eV). The processing route is contributing to the final color of the discs. The new route that includes three milling stages and SPS sintering is the most efficient in obtaining IR transparent glass-ceramics. The structural and optical analysis, taking into account also the aging of the samples in the ambient atmosphere, show promising results. Through future improvements, the doped partially amorphous ceramic, GZSe-3, is expected to be used in IR optical elements. It was found that the addition of barium and iodine in the Ge-Zn-Se alloy is a key aspect that influences the amount and stability of the amorphous phase in the glass-ceramic, but also the processing temperature. The optimization of the doping process is necessary. This study is an important step towards new IR transparent chalcogenides as bulks and thin films.

## Figures and Tables

**Figure 1 materials-15-05002-f001:**
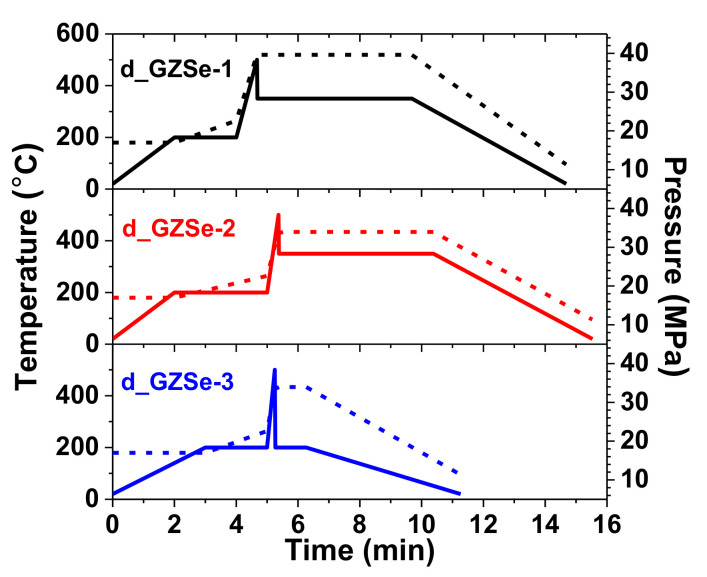
SPS sintering process. The temperature (solid lines) and applied pressure (dashed lines) used to densify the powders into bulk disk.

**Figure 2 materials-15-05002-f002:**
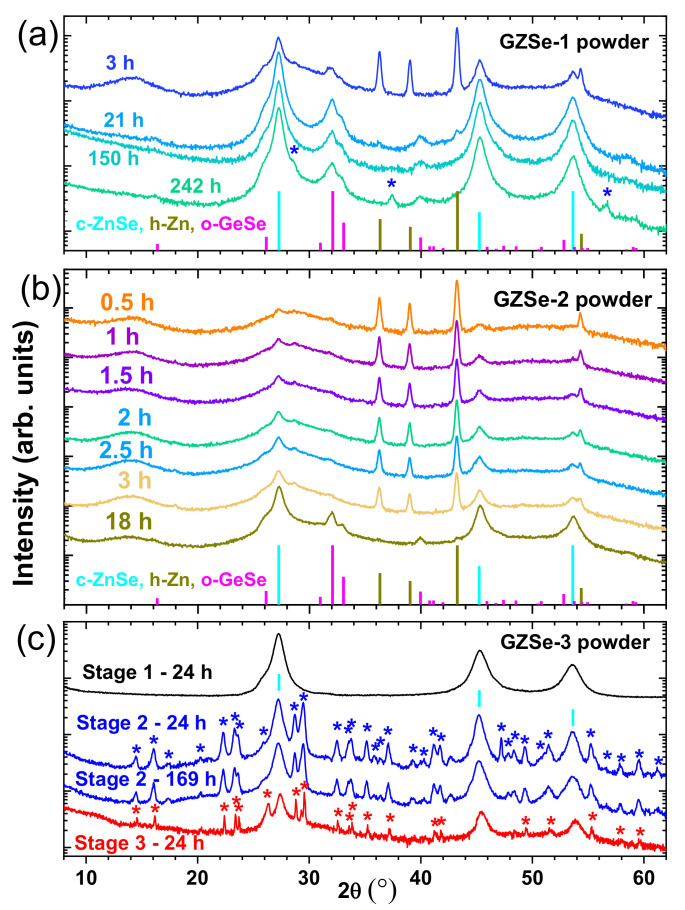
X-ray diffraction patterns of (**a**) GZSe-1, (**b**) GZSe-2, and (**c**) GZSe-3 powders taken at different milling times. The intensity is represented in logarithmic scale to make more visible the low intensity peaks, while the XRD patterns are expanded on the y-axis scale for clarity. The peaks of the unidentified phases are indicated with “*”.

**Figure 3 materials-15-05002-f003:**
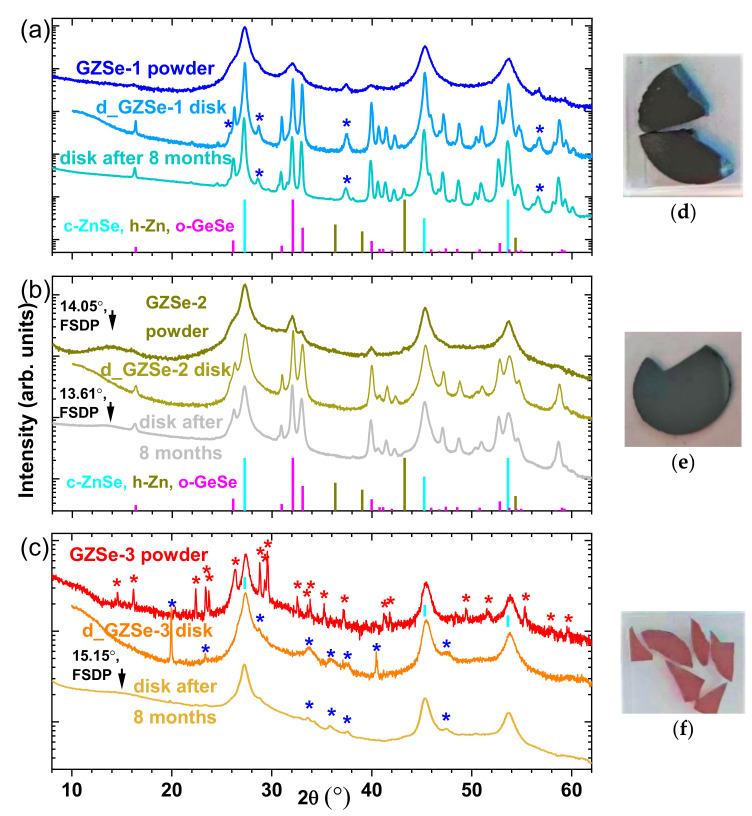
X-ray diffraction patterns of the sintered chalcogenide samples (**a**) d_GZSe-1, (**b**) d_GZSe-2, and (**c**) d_GZSe-3 obtained by Spark Plasma Sintering method, compared to those of the powders from which they were fabricated. The XRD measurements on bulk samples were repeated after 8 months of storage in the ambient atmosphere. The intensity is represented in logarithmic scale to make more visible the low intensity peaks, and the XRD patterns are expanded on the y-axis scale for clarity. The peaks of the unidentified phases were indicated with “*”. The optical images of the polished disk fragments (**d**) d_GZSe-1, (**e**) d_GZSe-2, and (**f**) d_GZSe-3.

**Figure 4 materials-15-05002-f004:**
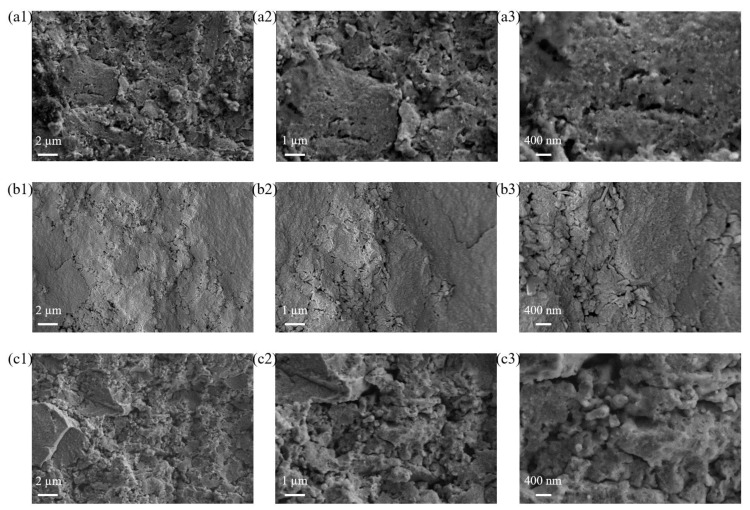
SEM images of the sintered samples: (**a1**–**a3**) d_GZSe-1; (**b1**–**b3**) d_GZSe-2; (**c1**–**c3**) d_GZSe-3. Magnifications are 5000× (1), 10,000× (2), and 20,000× (3) respectively. The scale bars are 2, 1, and 0.4 μm, respectively.

**Figure 5 materials-15-05002-f005:**
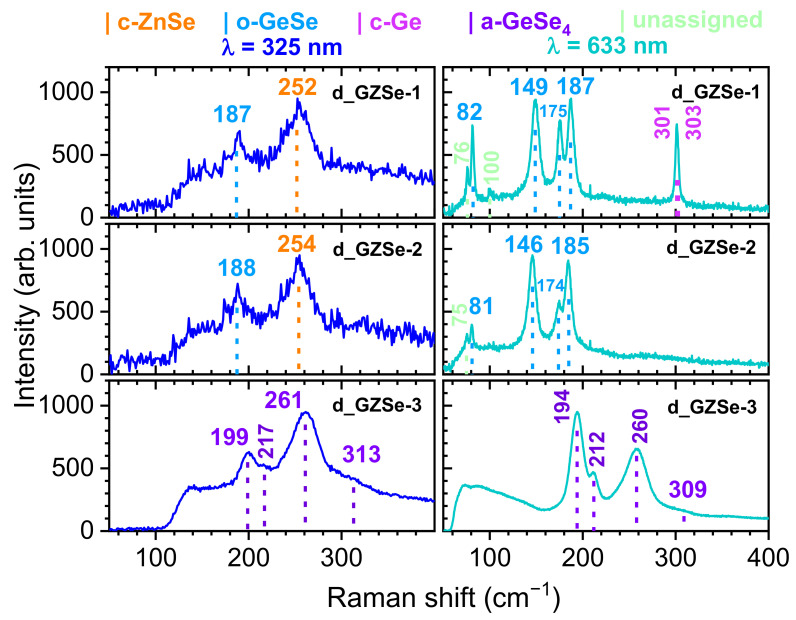
Micro-Raman spectra for the three bulk samples, when irradiated with monochromatic radiation with a wavelength of 325 nm (**left**) and 633 nm (**right**), respectively.

**Figure 6 materials-15-05002-f006:**
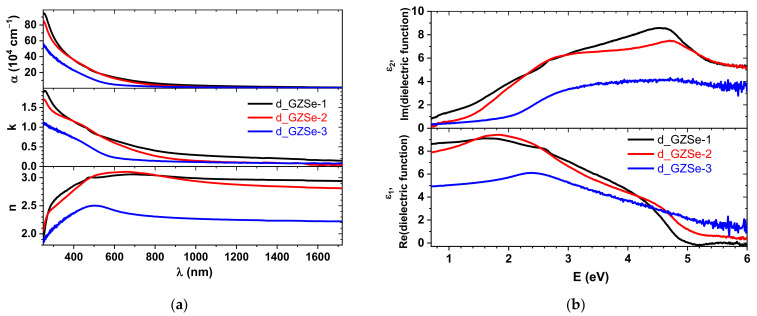
The optical properties of the chalcogenide bulk sintered samples on the mirror-polished surface: (**a**) the refractive index (n), extinction coefficient (k) and the absorption coefficient (α = 4πk/λ); (**b**) the dielectric function estimated6 from spectroscopic ellipsometry.

**Figure 7 materials-15-05002-f007:**
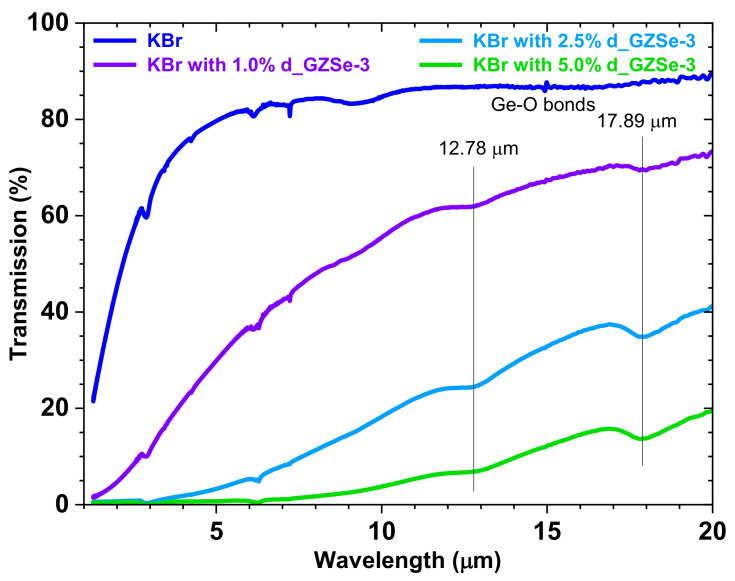
Transmission spectra of the KBr pellets with 1.0, 2.5 and 5.0% powder of d_GZSe-3.

**Table 1 materials-15-05002-t001:** The mechanical alloying synthesis parameters of the glass-ceramics compositions. The total milling time includes the pause.

Abbrev. Name	Composition (at. %), Composition (wt. %)	Raw Powders		WC Milling Balls	Powder’s Mass (g)	Total Milling Time (hours) / Speed (rpm)	Milling Time (min) / Pause (min)
		Substance	Mass (g) / at. %	Diameter (cm)/ Number			
GZSe-1	Ge_23.3_Zn_30.0_Se_46.7_, Ge_23.1_Zn_26.7_Se_50.2_,	GeSe_2_	12.4000 / 70	20 / 2	16.9225	150 / 400	3 / 6
		Zn	4.5225 / 30	10 / 5		92 / 400	3 / 3
GZSe-2	Ge_26.7_Zn_20.0_Se_53.3_, Ge_26.0_Zn_17.5_Se_56.5_,	GeSe_2_	12.4000 / 80	20 / 2	15.0381	18 / 400	3 / 6
		Zn	2.6381 / 20	10 / 5			
GZSe-3	*Stage 1*: Zn_50.0_Se_50.0_, Zn_45.3_Se_54.7_	Se	8.2050/50	20 / 2 15 / 2 10 / 10 5 / 30	15.0	24 / 400	3 / 6
		Zn	6.7950 / 50				
	*Stage 2*: Ba_6.2_Zn_26.6_Se_54.7_I_12.5_, Ba_10.1_Zn_20.4_Se_50.8_I_18.7_	ZnSe	4.4856 / 53.13		9.9433	169/400	3 / 3
		BaI_2_	2.8597 / 18.75				
		Se	2.5980 / 28.12				
	*Stage 3*: Ba_4.0_Ge_12.0_Zn_17.0_Se_59.0_I_8__.0_, Ba_6.7_Ge_10.6_Zn_13.5_Se_56.8_I_12.4_,	Ba_6.2_Zn_26.6_Se_54.7_I_12.5_ GeSe_2_	7.2303 / 64 3.6770 / 6		10.9073	24 / 400	3 / 3

**Table 2 materials-15-05002-t002:** The weight percentages of the phases formed in powders and in bulks as inferred from XRD. In the GZSe-3 samples, unidentified amorphous and crystalline phases, denoted with “*”, are present.

	GZSe-1, Ge_23.1_Zn_26.7_Se_50.2_ (wt. %)		GZSe-2, Ge_26.0_Zn_17.5_Se_56.5_ (wt. %)		GZSe-3, Ba_6.7_Ge_10.6_Zn_13.5_Se_56.8_I_12.4_ (wt. %)	
	Powder (wt. %)	Bulk (wt. %)	Powder (wt. %)	Bulk (wt. %)	Powder (wt. %)	Bulk (wt. %)
Ratio of phases	27.7 a-Ge_16.7_Se_11.0 _ 58.4 c-ZnSe 13.4 o-GeSe 0.5 h-ZnSe	57.4 c-ZnSe 33.6 o-GeSe 1.5 h-ZnSe 6.5 c-Ge	51.4 a-Ge_21.2_Se_30.2 _ 38.4 c-ZnSe 10.1 o-GeSe 0.1 h-Zn	19.7 a-Ge_2.1_Zn_5.2_Se_12.4_ 43.0 c-Zn_0.5_Ge_0.25_Se 37.3 o-GeSe	50.5 a- * 34.6 c-Zn_0.5_Ge_0.25_Se 14.9 c- *	51.4 a- * 43.4 c-Zn_0.5_Ge_0.25_Se 5.2 c- *

**Table 3 materials-15-05002-t003:** The average size of the crystallites for the phases formed in the powders and in the sintered glass-ceramics, as-prepared and after 8 months of Ambiental storage (aging).

	The Average Size of the Crystallites (nm)					
	c-ZnSe/c-Zn_0.5_Ge_0.25_Se			o-GeSe		
	Powder	Disk, as-Prepared	Disk, after 8 Months	Powder	Disk, as-Prepared	Disk, after 8 Months
d_GZSe-1	10.8	33.1	57.7	13.9	53.3	119.9
d_GZSe-2	9.5	10.0	13.3	16.5	27.0	56.4
d_GZSe-3	13.4	6.8	7.6	-	-	-

**Table 4 materials-15-05002-t004:** The elemental EDS composition (at. %) of the sintered samples as compared with the starting powders. In addition, the computed weight percentages of the crystalline phases of the bulk samples inferred from XRD are included for comparison (the unidentified amorphous and crystalline phases are denoted with “*”).

Disk’s Abbreviated Name	Composition of Sintered Samples from EDS Spectra (at. %)	The Atomic Percentages in the Starting Powders	The Weight Percentages of the Phases Formed in Bulks as Inferred From XRD
d_GZSe-1	Ge_26_Zn_32_Se_42_	Ge_23.3_Zn_30.0_Se_46.7_	57.4% c-ZnSe; 33.6% o-GeSe; 1.5% h-ZnSe; 6.5% c-Ge
d_GZSe-2	Ge_28_Zn_21_Se_51_	Ge_26.7_Zn_20.0_Se_53.3_	19.7% a-Ge_2.1_Zn_5.2_Se_12.4_; 43.0% c-Zn_0.5_Ge_0.25_Se; 37.3% o-GeSe
d_GZSe-3	Ba_2_Ge_16_Zn_19_Se_60_I_3_	Ba_4.0_Ge_12.0_Zn_17.0_Se_59.0_I_8.0_	51.4% a- *; 43.4% c-Zn_0.5_Ge_0.25_Se; 5.2% c- *

**Table 5 materials-15-05002-t005:** The peaks positions in the Raman spectra for the three mirror-polished chalcogenide sintered bulks, when irradiated with monochromatic radiation with a wavelength of 325 nm and 633 nm. The intended composition (where coefficients are in wt. %) and the percentage (wt. %) of each phase are also presented.

	d_GZSe-1		d_GZSe-2		d_GZSe-3	
	Ge_23.1_Zn_26.7_Se_50.2_		Ge_26.0_Zn_17.5_Se_56.5_		Ba_6.7_Ge_10.6_Zn_13.5_Se_56.8_I_12.4_	
	60.0% ZnSe 33.3% o-GeSe 6.7% c-Ge		19.7% a-Ge_1.9_Zn_7.7_Se_10.1_ 43.0% c-Zn_0.5_Ge_0.25_Se 37.3% o-GeSe		51.4% a-* 43.4% c-Zn_0.5_Ge_0.25_Se 5.2% c-*	
	**λ = 325 nm**	**λ = 633 nm**	**λ = 325 nm**	**λ = 633 nm**	**λ = 325 nm**	**λ = 633 nm**
**Raman peaks (cm^−1^)**		76		75		
		82 (GeSe)		81 (GeSe)		
		100				
		149 (GeSe)		146 (GeSe)		
		175 (GeSe)		174 (GeSe)		
	187 (GeSe)	187 (GeSe)	188 (GeSe)	185 (GeSe)	199 (a-GeSe_4_)	194 (a-GeSe_4_)
	252 (ZnSe)		254 (ZnSe)		217 (a-GeSe_4_)	212 (a-GeSe_4_)
		301 (Ge)			261 (a-GeSe_4_)	260 (a-GeSe_4_)
		303 (Ge)			313 (a-GeSe_4_)	309 (a-GeSe_4_)

**Table 6 materials-15-05002-t006:** The peaks positions in the Raman spectra for materials indicated in other studies. The bolded values are for the most intense peaks in each spectrum. TA, transverse acoustic mode; LA, longitudinal acoustic mode; TO, transverse optical phonon mode; LO, longitudinal optical phonon mode; LO(Γ), LO at k = 0 (Γ point); LA(X), LA at the point X (1, 0, 0); CS, corner-sharing GeSe_4/2_ tetrahedra; ES, edge-sharing Ge_2_Se_8/2_ bi-tetrahedra.

	GeSe	ZnSe		Ge	a-GeSe_4_
	λ = 633 nm	λ = 532 nm	Inelastic Neutron Scattering	λ = 633 / 514.5 nm	λ = 785 nm
**Raman** **peaks** **(cm^−1^)**	A_g_^3^ 82 B_3g_^1^ **151** A_g_^2^ 175 A_g_^1^ 188 [19]	2TA(X) 139 TO(Γ) 204 LO(Γ) **251** [20]	TA(X) 70 LA(X) 194 TO(Γ) 213 LO(Γ) **253** [21]	E_2_ 303.3 [22] **/** 300.5 [23]	CS **195** [24,25] ES 211 [24], 213 [25] polymeric Se_n_ chains: 257 [24], 259 [25] ES 298 [24], 285 [25]

## Data Availability

The data presented in this study are available on a reasonable request from the corresponding author.

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
