# Peer review of "New Chalcogenide Glass-Ceramics Based on Ge-Zn-Se for IR Applications"

_materials, 2022, doi:10.3390/ma15145002_

Round 1

Reviewer 1 Report

·       English language  should be improved and revised using experts

·       The abstract should contain some of the main results to express the value of the work. Therefore, the authors should re-write the abstract by adding some informative results.

·       The introduction, the authors, wrote, “ These IR devices have two functionalities: night vision (in absolute dark) of bodies or objects at room temperature, which emit IR radiation between 7 ÷ 14 μm and thermovision of warm bodies or objects, which emit IR radiation between 3 ÷ 5 μm. The second functionality is very useful to detect patients with fever e.g. due to Covid infection.”   What are the authors mean by “ 7 ÷ 14 μm “ and  3 ÷ 5 μm

·       The last paragraph in the introduction section should be improved by adding an extra paragraph illustrating the work's objective and the novelty and the difference between the previous and current work.

Author Response

Dear Reviewer, thank you for your time and valuable comments. Below are our answers and text changes based on your suggestions.

  1. English language should be improved and revised using experts

Answer: The writing of the manuscript was reviewed by several fluent academic English colleagues and improved.

Text changes: All the corrections and language improvements are marked with “Track Changes” in the revised manuscript.

  1. The abstract should contain some of the main results to express the value of the work. Therefore, the authors should re-write the abstract by adding some informative results.

Answer: Before submission we have rewritten the abstract two times, however, it is difficult to highlight all the results due to the 200 words limit (according of the journal’s rules). We believe that most of the informative results, that are of importance for glass ceramic community, such as (i) the fact that the amount of the amorphous phase is 19.7, and 51.4 wt. % in two of the compositions (d_GZSe-2 and d_GZSe-3), while one is 100% polycrystalline (d_GZSe-1); (ii) the main crystalline phase is cubic ZnSe and orthorhombic GeSe for the first two prepared samples; (iii) doping with barium and iodine preserves the amorphous phase formed by milling and lowers the sintering temperature from 350 °C to 200 °C; (iv) the color of the first two sintered samples is black (band gap values are 0.42 and 0.79 eV), while that of the latter is red (Eg is 1.37 eV); and finally (v) the doped sintered sample (d_GZSe-3) is transparent in IR domain which was the goal of this study. The abstract was rewritten for clarity.

Text changes:

(page 1, rows 14-26) The consumer market requests infrared (IR) optical components, made of relatively abundant and environmentally friendly materials, to be integrated or attached to smartphones. For this purpose, three new chalcogenides samples namely Ge23.3Zn30.0Se46.7 (d_GZSe-1), Ge26.7Zn20.0Se53.3 (d_GZSe-2) and Ba4.0Ge12.0Zn17.0Se59.0I8.0 (d_GZSe-3) were obtained by mechanical alloying and processed by Spark Plasma Sintering into dense bulk disks. Obtaining a completely amorphous and homogeneous material proved to be difficult. d_GZSe-2 and d_GZSe-3 are glass-ceramics with the amount of the amorphous phase being 19.7 and 51.4 wt. %, while d_GZSe-1 is fully polycrystalline. Doping with barium and iodine preserves the amorphous phase formed by milling and lowers the sintering temperature from 350 °C to 200 °C. The main crystalline phase in all the prepared samples is cubic ZnSe or cubic Zn0.5Ge0.25Se, while the amorphous phase is formed GeSe4 clusters in d_GZSe-3. The color of the first two sintered samples is black (the band gap values are 0.42 and 0.79 eV), while d_GZSe-3 is red (Eg is 1.37 eV) and is transparent in IR domain. These results are promising for future research in IR materials and thin films.

  1. The introduction, the authors, wrote, “ These IR devices have two functionalities: night vision (in absolute dark) of bodies or objects at room temperature, which emit IR radiation between 7 ÷ 14 μm and thermovision of warm bodies or objects, which emit IR radiation between 3 ÷ 5 μm. The second functionality is very useful to detect patients with fever e.g. due to Covid infection.” What are the authors mean by “ 7 ÷ 14 μm “ and 3 ÷ 5 μm

Answer: These values are the wavelengths for the IR atmospheric windows. Roughly, in these regions there is relatively low absorption of IR radiation by atmospheric gases and it can be detected.

Text changes:

(page 1, rows 35-40) These IR devices have two functionalities: (i) night vision of bodies or objects at room temperature which emit IR radiation with the wavelength between 7 ÷ 14 μm and (ii) thermovision of warm bodies or objects which emit IR radiation with the wavelength between 3 ÷ 5 μm. The second functionality is also very useful to detect patients with fever due infectious diseases, that can be easily transmitted leading to outbreaks.

  1. The last paragraph in the introduction section should be improved by adding an extra paragraph illustrating the work's objective and the novelty and the difference between the previous and current work.

Answer: We thank the reviewer for the suggestion. In the revised manuscript the last paragraph from introduction was rephrased to highlight the purpose of this work.

Text changes:

(page 2, rows 77-88) The objective of this work is to produce glass ceramics from the Ge-Zn-Se system, with the Zn concentration greater than the 5% limit, undoped or doped with Ba and I and transparent in the IR domain. For this purpose, a new preparation route involving mechanical alloying and SPS, is explored. New chalcogenide compositions, namely Ge23.3Zn30.0Se46.7, Ge26.7Zn20.0Se53.3 and Ba4.0Ge12.0Zn17.0Se59.0I8.0 were obtained by ball milling. Subsequently, the powders were processed by Spark Plasma Sintering (SPS) to produce the bulk glass-ceramics disks. The structural evolution of the bulk materials was characterized using both grazing incidence X-ray diffraction (GIXRD) and Raman spectroscopy and compared with the precursor powders. while its morphological changes were observed using both optical and scanning electron microscopy. Moreover, the optical properties of the obtained materials were investigated by spectroscopic ellipsometry and Fourier transform infrared (FTIR) spectroscopy.

Reviewer 2 Report

Velea et al. presented results on the synthesis and characterization of chalcogenide glass ceramics. The manuscript can be published in the journal after some revisions.

1. For GZSe-1 the composition is given to the third decimal place, for GZSe-2 to the second decimal place. The instrument allows to obtain matching data with an accuracy of up to three decimal places?

2. The language and style of the entire text should be reviewed.

3. I suggest that the GIXRD data and the chemical composition of the samples obtained be combined into a table for the reader's convenience.

4. In my opinion, it would be useful to briefly discuss why the color of GZSe-3 spices depends on the grinding time.

Author Response

Dear Reviewer, thank you for your time and valuable suggestions. Below are our answers and text changes based on your comments.

  1. For GZSe-1 the composition is given to the third decimal place, for GZSe-2 to the second decimal place. The instrument allows to obtain matching data with an accuracy of up to three decimal places?

Answer: In the revised manuscript, all the composition (at. % or wt. %) are given with a precision of one number after the decimal point.

Text changes: All the corrections are marked with “Track Changes”.

  1. The language and style of the entire text should be reviewed.

Answer: In the revised manuscript, the English language and style of the entire text was improved using an expert colleague.

Text changes: All the revisions and language improvements are marked with “Track Changes” in the revised manuscript.

  1. I suggest that the GIXRD data and the chemical composition of the samples obtained be combined into a table for the reader's convenience.

Answer: In Table 4 a new column was added. The chemical composition of the starting powders inferred from EDS, can be easily compared with the atomic composition of the starting powders and phase percentages measured by XRD on bulk samples.

Text changes:

(pages 10-11, row 350)

Table 4. The elemental EDS composition (at. %) of the sintered samples as compared with the starting powders. In addition, the computed weight percentages of the crystalline phases of the bulk samples inferred from XRD are included for comparison.

Disk’s abbreviated name

The composition from EDS spectra

The atomic percentages in the starting powders

The weight percentages of the phases formed in bulks as inferred from XRD

d_GZSe-1

Ge26Zn32Se42

Ge23.3Zn30.0Se46.7

57.4 % c-ZnSe; 33.6 % o-GeSe; 1.5 % h-ZnSe; 6.5 % c-Ge

d_GZSe-2

Ge28Zn21Se51

Ge26.7Zn20.0Se53.3

19.7 % a-Ge2.1Zn5.2Se12.4; 43.0 % c-Zn0.5Ge0.25Se; 37.3 % o-GeSe

d_GZSe-3

Ba2Ge16Zn19Se60I3

Ba4.0Ge12.0Zn17.0Se59.0I8.0

51.4 % a-*; 43.4 % c- Zn0.5Ge0.25Se; 5.2 % c-*

  1. In my opinion, it would be useful to briefly discuss why the color of GZSe-3 spices depends on the grinding time.

Answer: Thank you for suggestion! The color of GZSe-3 powder does not only depend on the milling time but on all the sintering conditions. A new paragraph was added in the Discussion section of the revised manuscript trying to explain the reason behind GZSe-3 light red color.

Text changes:

(page 15, rows 491 – 501) The GZSe-3 powder was obtained at the end of three successive sintering stages, and the color of the powders obtained at the end of these stages is intriguing. Thus, after the first stage, the ZnSe powder is black, a result also obtained in other works [35]. This shows that the surface of the ZnSe nanometric sized particles is very defective, and given the large number of surface atoms relative to the volume atoms [35], these defects drastically decrease the value of the bandgap, leading to a dark colored powder. At the end of the second stage, the addition of BaI2 and Se to ZnSe produced a dark red Ba6.2Zn26.6Se54.7I12.5 powder. Finally, the incorporation of GeSe2 causes the final GZSe-3 powder to become light red. This might be due to the formation of the amorphous GeSe4 (which is red [33]) in the GZSe-3 sample, as the Raman Spectroscopy revealed. Further sintering by SPS did not produce any changes in the color of the d_ GZSe-3 bulk disc.

Reviewer 3 Report

Please define all the acronyms before their first appearance in text “IR optical “..and so on. Everywhere it was found the same

Please check the entire paper together with a native English speaker cause there are many grammar mistakes

Some sentence are inconsistent “The doped sintered sample is transparent” which sample do you speak about ?

“The second functionality is very useful to detect patients with fever e.g. due to Covid infection” OK, I see your point but this disease will disappear probably soon. You have to formulate in more professional way !

The state of art if very briefly presented and without a proper structure – please revise it

Which is actually the scientific novelty of this work ?

The same ratios are maintained after 150 hours of milling (XRD data are not shown).” You can put this as an appendix !

The scale bar is not well visible and Figure 4

Some quantitative results are required in conclusion to link with your results section

Most of references are out of date- therefore more recent one are required

Author Response

Dear Reviewer, thank you for your time and valuable suggestions. Attached are our answers and text changes based on your comments.

Round 2

Reviewer 3 Report

.